# OpenReview forum: "Instruction Following by Boosting Attention of Large Language Models"
_ICLR.cc/2026/Conference — Submitted to ICLR 2026_

### Official Review · Reviewer_psxi · 2025-10-21

**Soundness:** 1
**Presentation:** 2
**Contribution:** 1
**Rating:** 2
**Confidence:** 5

**Summary:**

The authors propose a unified benchmark for attention control methods, along with a new attention-steering algorithm called InstaBoost. However, the paper falls short of its goals. The proposed benchmark combines existing, well-known datasets, and the InstaBoost method closely mirrors the previously introduced **PASTA** algorithm, with only minor modifications on the selection of heads.

**Strengths:**

- **Simplicity:** The proposed attention steering method is straightforward to implement, requiring no additional training and can be applied easily with off-the-shelf code.

**Weaknesses:**

**Benchmarking:**
The benchmark proposed by the authors primarily consists of commonly used datasets already present in prior work (as stated in Page 3, Section 2.1). By examining Appendix A.4,  we can see that it includes datasets such as *TriviaQA* and *TruthfulQA*, among others. Therefore, the benchmark adds little novelty or substantial value to justify its contribution.

**Methodology:**
Given the limited novelty of the benchmark, the primary contribution revolves around the InstaBoost algorithm. However, InstaBoost replicates the PASTA approach while differing only in minor implementation details. Parituclarly, the authors follow Spotlight in terms of selection of all heads. Apart from that, there are no key differentiations with PASTA. Hence, the methodological contribution is also minimal.

**Experiments:**
1. The experiments are conducted solely on Meta-LLaMA model, without validation across other model architectures. Since attention mechanisms vary across models (e.g., full vs. sparse or windowed attention), broader experimentation is essential to confirm generalizability of the method and investigation of potential pitfalls.
2. In Appendix A.3, PASTA was tested using a single hyperparameter setting, while InstaBoost was tuned over an interval and reported with its best results. This inconsistency raises concerns about the fairness of the comparison.

**Minor Comments:**
When describing the transformer layers and their internal mechanisms, the authors overlook the role of residual connections, which are fundamental parts of these models. For rigor, this aspect should be explicitly included in the methodology description.

**Questions:**

Please refer to the points outlined in the Weaknesses section.

---

### Official Review · Reviewer_PyUW · 2025-10-27

**Soundness:** 2
**Presentation:** 3
**Contribution:** 2
**Rating:** 4
**Confidence:** 4

**Summary:**

This paper studies how to steer language models to follow user-specified constraints or to control generation along a desired attribute. It compares three families of low-resource interventions: prompt-based methods that modify only the input, latent-space methods that add or project vectors in hidden states, and attention-based methods that reweight attention over instruction vs user tokens. The authors first benchmark prompt vs latent steering across 15 behavioral tasks and show that which one works better is task-dependent. They then introduce attention steering, review PASTA and Spotlight, and propose InstABoost, which boosts post-softmax attention on instruction tokens for all heads. InstABoost performs on par with or better than the baselines on many tasks. In the side-effects analysis, increasing strength with InstABoost preserves fluency and relevance better than alternatives.

**Strengths:**

- Provides a useful comparative study of prompt, latent, and attention steering across a broad set of tasks.
- The proposed method performs on par with prior work and appears more robust to hyperparameter changes.
- Multi-objective evaluation: reports steering success alongside fluency and relevance, with strength sweeps that expose trade-offs.

**Weaknesses:**

- The proposed method feels like a small variation on PASTA and SPOTLIGHT, and the motivation for preferring a multiplicative boost over an additive adjustment is not well developed. The paper should explain why the chosen form is expected to work better or be more robust.
- PASTA selects specific heads to modify. What happens if one applies the same uniform rescaling across all heads, as done in SPOTLIGHT and in the proposed method? Would performance degrade, or would it be comparable? An ablation here would clarify whether head selection is essential.
- The paper could better situate its study within the broader steering literature. It briefly mentions AXBench, but a more detailed comparison and discussion would help readers understand how the findings relate to existing evaluations of latent steering.

**Questions:**

1. What is the motivation for choosing a multiplicative post-softmax boost over an additive adjustment to attention scores? Under what conditions should the multiplicative form outperform or be more robust than the additive one?
2. What happens if you apply the same rescaling uniformly across all heads, as in Spotlight and your method? Can you report an ablation comparing head selection vs uniform all-heads for PASTA?

---

### Official Review · Reviewer_p32r · 2025-10-30

**Soundness:** 2
**Presentation:** 1
**Contribution:** 1
**Rating:** 4
**Confidence:** 3

**Summary:**

This paper proposes a new method called InstABoost, which achieves behavioral control by multiplicatively enhancing attention to instruction tokens. The authors construct a benchmark comprising 15 tasks and demonstrate experimentally that InstABoost outperforms or matches the best existing methods on most tasks while better preserving generation quality. This work establishes attention guidance as an efficient and reliable paradigm for behavior control.

**Strengths:**

1. The paper introduces a novel method, INSTABOOST, that is remarkably simple and efficient, requiring no costly head-selection process.
2. It establishes a unified benchmark consisting of 15 diverse behavioral control tasks.

**Weaknesses:**

1. Although the constant amplification factor in InstABoost simplifies implementation, it may not be optimal. The dynamic nature of attention distribution demands finer control (e.g., context-adaptive scaling), otherwise it may cause over-focusing or neglect of key information under complex inputs.

2. The paper lacks a thorough theoretical explanation of the design rationale behind the model, making the work appear more like an engineering enhancement rather than a methodological innovation.

3. Despite claims of improving controllability, the paper does not systematically evaluate InstABoost on safety-related tasks such as adversarial robustness, hallucination mitigation, or moral alignment.

4. While avoiding head selection, the paper does not sufficiently explain how hyperparameters are chosen or transferred across tasks, leaving the tuning process potentially costly.

**Questions:**

1. If the model size increases, would the constant amplification factor remain applicable? Would its value need to be adjusted?
2. In production environments, what are the inference latency and memory overheads of INSTABOOST (especially for large models or long contexts)? How do these compare quantitatively with baselines (no-steer, Spotlight, PASTA)?
3. Since Gemini 2.0 is used for fluency/relevance evaluation, how do the authors control for potential biases introduced by this model? Was any human evaluation conducted to validate the reliability of the automatic scores?
4. Have the authors experimented with adaptive or learned amplification factors across layers, heads, or tokens (e.g., based on attention entropy, gradient information, or lightweight adapters)? If so, what were the results? If not, do they consider this the most direct direction for improvement?
5. Does hyperparameter selection depend on the task or dataset? Is there a default configuration that generalizes across tasks? Could the authors provide tuning guidelines or a quantitative sensitivity analysis?
6. Could the authors present typical failure cases (types of instructions or inputs where INSTABOOST fails or degrades significantly)? This would help clarify the method’s boundary conditions.

---

### Official Review · Reviewer_jjgD · 2025-11-01

**Soundness:** 2
**Presentation:** 3
**Contribution:** 2
**Rating:** 2
**Confidence:** 4

**Summary:**

The paper proposes INSTABOOST, a very simple inference-time attention steering rule that multiplies post-softmax attention on instruction tokens and renormalizes, applied to all heads and layers. The authors also assemble a “behavior control” benchmark across 15 tasks drawn from emotions, personas, jailbreaks, toxicity, truthfulness, and general QA, and compare prompting, several latent-steering baselines, and attention steering methods PASTA and Spotlight. They report that INSTABOOST is competitive or best on most tasks and avoids some side effects seen with PASTA and Spotlight.

**Strengths:**

1. Simplicity and practicality - The method is a one-line multiplicative boost on post-softmax attention to the instruction segment. It needs no head profiling or per-task search

2. Empirical competitiveness on their suite - On the 15-task benchmark, INSTABOOST is competitive, including difficult jailbreak settings where other attention methods stall. The per-family averages in Table 1 show strong and stable behavior for attention methods in general, with INSTABOOST the only attention method that does well on jailbreak.

**Weaknesses:**

1. Limited novelty - Compared to PASTA and Spotlight, INSTABOOST is essentially  a multiply post-softmax attention on the instruction tokens by a const everywhere. The idea of reallocating attention mass toward marked spans is known, and the paper’s main change is multiplicative post-softmax scaling across all heads rather than per-head post-softmax downweighting (PASTA) or additive pre-softmax targeting (Spotlight). The paper acknowledges these predecessors and positions INSTABOOST as a simpler trade-off but this is still incremental.

2. Evaluation tasks focus a lot on execution, not complex instruction following - Many reported results are on emotion or persona shaping, toxicity, truthfulness, and QA. These are useful, but they are not classic instruction-following stress tests like IFEval, IFBench, ComplexBench that directly score instruction adherence independent of world knowledge or task difficulty. A steering for instruction following paper should include at least one instruction-centric benchmark to isolate steering quality from task success. The current setup may over-credit methods that help general task execution.

3. LLM-as-judge without reliability checks - Fluency and relevance are scored by Gemini 2.0 Flash with fixed prompts. There is no inter-rater agreement, rubric validation, or human calibration. Prior work shows coarse LLM judging can be biased by phrasing and style [1,2], which is especially risky here because attention steering directly changes phrasing. Fine-grained response content evaluations as in [3] would be better.

4. Why boosting helps is not well explained mechanistically - The motivation cites work showing rule following drops when attention to rules is reduced, then assumes the inverse should help. The paper does not probe where in depth the improvement arises, for example which layers contribute most, whether gains concentrate in a small head subset, or whether boosting mainly combats under-allocation at late layers.

5. Instruction span identification is underspecified - The dscriptions in method section assume specified instruction tokens. In real multi-turn or tool-use prompts these are not pre-specified. The paper should spell out how instruction tokens are identified for boosting.

6. Single-model scope - All results use Llama-3-8B-Instruct. Claims about robustness across tasks would be stronger with a second backbone and size.

References:

[1] Zheng et al., (2023). Judging LLM-as-a-Judge with MT-Bench and Chatbot Arena.

[2] Chen et al., (2024). Humans or LLMs as the Judge? A Study on Judgement Bias.

[3] Stolfo et al., (2025). Improving Instruction-Following in Language Models through Activation Steering.

**Questions:**

1. Steering metric validity - How exactly is “steering success” computed for each task and how comparable are those numbers across tasks? Any human calibration or agreement studies for the Gemini-based fluency and relevance scores?

2. Instruction tokens identification - For each dataset, how is the instruction span determined when prompts include system text, user queries, few-shot demos, or multi-turn context? Any failures when instructions are repeated or fragmented?

3. Where does the gain come from - Please add ablations on: boost only early vs only middle vs only late layers. Also boost only a subset of heads vs all heads. Report which layers and heads account for most of the win. This would turn the mechanism from a cartoon into evidence.

4. Hyperparameters. Provide a table of exact constants and hyperparams per task, grid ranges, and the selection criterion used when trading off steering success vs fluency vs relevance.

5. Add an instruction-following benchmark. Include IFEval-style tests or an equivalent fine-grained instruction suite to measure steering fidelity directly, separate from knowledge and task difficulty.

---

### Meta-Review · Area_Chair_9zrw · 2026-01-07

**Summary:**

The reviewers unanimously found that the method offers limited novelty over existing approaches (PASTA and Spotlight). Moreover, the experimental comparison was regarded as unfair because the proposed method was extensively tuned while baselines were not.

**Reviewer Concerns:**

The authors did not submit rebuttals and none of the concerns have been addressed.

**Reviewer Scores:**

The authors did not submit rebuttals. Therefore, the reviewer scores would not been changed due to this.

---

### Decision · Program_Chairs · 2026-01-26

Reject